# Willingness and Influencing Factors of Pig Farmers to Adopt Internet of Things Technology in Food Traceability

**Ruiyu Sun [1], Siyao Zhang [1], Tianyu Wang [1], Jiarui Hu [1], Junhu Ruan [1]** 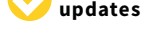 **and Junyong Ruan [2,*]**

[1]   College of Economics and Management, Northwest A&F University, Xianyang 712100, China;
sunruiyu@nwafu.edu.cn (R.S.); jiayouzsy@nwafu.edu.cn (S.Z.); wty@nwsuaf.edu.cn (T.W.);
hjrljgxy@nwafu.edu.cn (J.H.); rjh@nwsuaf.edu.cn (J.R.)
[2]   Medical Engineering Department, Qingdao Special Servicemen Recuperation Center of PLA Navy,
Qingdao 266071, China
*   Correspondence: ruanjunyongqdyl@163.com

**Abstract:** The Internet of Things technology (IoT) in food traceability provides new ideas to solve the problem of smart production and offers new ideas for the formation of safe and high-quality markets for meat products. However, scholars have studied the combination of blockchain and IoT technology. There is a lack of research on the combination of IoT and food traceability technology. Moreover, previous studies focused on the application of IoT traceability technology, taking farmers' adoption willingness as an exogenous variable while ignoring its endogeneity. Therefore, it is essential to study farmers' willingness to adopt IoT traceability technology and find the factors that influence farmers' adoption intention. Based on survey data from 264 pig farmers in Shaanxi Province, this paper discussed the factors which influence pig farmers' adoption of the technology by using the Unified Theory of Acceptance and Use of Technology (UTAUT). The results showed that farmers' adoption intention was influenced by a combination of farmers' performance expectancy, effort expectancy, social influence, personal innovation, and perceived risk. Personal innovation played a mediating role in effort expectancy and adoption willingness and perceived risk played a moderating role in personal innovation and adoption willingness.

**Keywords:** pig farmers; adoption willingness of IoT traceability technology; Unified Theory of Acceptance and Use of Technology; Latent Moderate Structural Equations

## 1. Introduction

With the development of the economy, people's quality of life is constantly improving, and the proportion of meat food in people's daily dietary needs is increasingly high. At the same time, the transmission of COVID-19 has created unquantifiable damage. The economy has been destroyed by this virus and immediate action is required. D'Adamo et al. [1] pointed out that the availability of infrastructure was necessary to generate economic growth and social opportunities without compromising environmental protection. Moreover, infrastructure could influence, directly or indirectly, about 72% of the targets in terms of the Sustainable Development Goals [2]. Following this approach, D'Adamo et al. [1] suggested that favoring digitalization could be implemented in order to improve our lives. Therefore, the application of IoT traceability technology in food as a digital infrastructure should be emphasized.

Traceability is the ability to follow the movement of food products throughout food supply chains [3]. When people find that there are quality or safety issues in food, they can locate the problem and in turn the cause based on the product traceability system. The IoT traceability technology refers to the technology that realizes the function of food traceability through the Internet of Things. Through the use of the Internet of Things and various sensors, such as the global positioning system (GPS), geographic information system (GIS), near-field communication (NFC), radio frequency identification (RFID) and

temperature and humidity sensors, monitoring and information capturing can be improved in various processes, such as production, processing, storage, distribution, and retail [4]. However, due to the cost of applying IoT traceability technology, farmers' perception, technology acceptance and production privacy, farmers' willingness to adopt IoT traceability technology is different. Therefore, it is of great practical significance to analyze farmers' willingness and influencing factors to adopt IoT traceability technology and identify the key influencing factors for solving the food safety problems facing China and connecting farmers to the modernized large market.

China has a huge pork market and is the largest pork producer in the world. Affected by the African swine fever epidemic, the proportion of pork in the total meat market has dropped sharply. However, based on past consumption habits, there is still more space for callback in the pork market in the future [5]. The application of IoT technology in pig farming can reduce labor costs and improve production efficiency, which is of great importance in large-scale pig production [6]. Therefore, this study selected pig farmers as the subjects to illustrate the influencing factors of pig farmers' adoption willingness of IoT traceability technology, and accordingly proposes policy recommendations, which are important for further promoting the application of IoT traceability technology in pig farming.

Existing research on IoT traceability technologies is mainly characterized by the following.

Firstly, most scholars have studied blockchain, IoT technology, and the combination of blockchain and traceability technology. For example, Reyna et al. [7] pointed out that blockchain could enrich the IoT by providing a trusted sharing service, where information was reliable and could be traceable. Data sources could be identified at any time and data remained immutable over time, which increased its security. Therefore, the use of blockchain could complement the IoT with reliable and secure information. It is considered that in future research, the block structure should be studied to improve data retrieval efficiency by combining the characteristics of IoT engineering. Previous studies explored the methods of using blockchain for traceability system construction in various daily food and dual-use foods, and explored the deeper promotion role of block chain technology in food traceability system construction. Furthermore, Kamilaris et al. [8] proposed that blockchain was a promising technology towards a transparent supply chain of food, but many barriers and challenges still existed, which hindered its wider popularity among farmers and food supply systems. The challenges involved accessibility, governance, technical aspects, policies, and regulatory frameworks.

Secondly, existing studies focused on the application of IoT traceability technology, taking farmers' adoption willingness as an exogenous variable while ignoring its endogeneity. For example, Ma Peng [9] proposed several methods and measures for the construction of the traceability system of plateau summer vegetables based on IoT technology by combining the current situation of planting and sales of a variety of summer vegetable agricultural products enterprises, such as agricultural cooperatives and plateau summer vegetable sales enterprises in Yuzhong County, Lanzhou City. Moreover, based on a practical application case, which was a city's Food and Drug Administration using quick response code (QR code), integrated circuit card (IC card), and traceability code as the carrier to collect and record the traceability information of each link for the construction of the city's food safety traceability system, and realized the complete information traceability of food in production, circulation, storage, and consumption links, researchers proposed that with the support of information technology such as cloud computing, big data and mobile Internet, the core of improving supervision efficiency, and the implementation of the main responsibility of food safety of production operators as the landing point, the government should take the lead and enterprises should be responsible for establishing a scientific, complete, and efficient food safety traceability system in order to fully protect food safety for the general public.

In summary, through the collation of existing studies, it was found that scholars have made great academic achievements in the study of IoT traceability technology, which has

important theoretical reference value for this study. However, there is still a need for improvement in at least the following aspects. Firstly, as the previous literature mainly studied the combination of IoT and blockchain, there is a lack of research on applying IoT to food traceability. Therefore, it is necessary to study the combination of IoT and traceability technology, especially food traceability technology. Secondly, most of the literature on IoT traceability technology ignored farmers' adoption intention which is an important endogenous variable, so we should emphasize the influence of farmers' adoption willingness of IoT traceability technology in promotion and application of this technology. Furthermore, Jurgilevich et al. [10] summarized that the European Union Commission has identified three main stages of the food system with reference to the circular economy: production, consumption, and waste. The research of farmers' willingness to adopt the technology is to ensure the circularity of food system in the production stage. As mentioned above, this paper takes Shaanxi Province as an example and explores the factors influencing pig farmers' willingness to adopt IoT traceability technology from the microscopic perspective of pig farmers based on the innovative Unified Theory of Acceptance and Use of Technology (UTAUT) model, and provides a theoretical basis for the formulation of relevant policies to increase the popularity of IoT traceability technology in rural areas.

## 2. Theoretical Models and Research Hypotheses

### 2.1. Theoretical Model

The Technology Acceptance Model (TAM) was mostly used in previous studies. The TAM model was proposed by Davis et al. (1989) based on the Theory of Reasoned Action (TRA model) with reference to self-efficacy theory, input-output theory, and other related theories. The TAM model was mainly used to predict and explain users' perceived acceptance of a new information system after using it for a period of time, the purpose of which was to find out the reasons why people accepted or rejected new information systems. The TAM model assumes that the actual usage behavior of users for a specific information system in an organization is determined by their usage intention, which is determined by both users' usage attitude and perceived usefulness, while users' usage attitude will be determined by both users' perceived usefulness and perceived ease of use [11]. Finally, users' perceived usefulness and perceived ease of use are influenced by external factors. The external factors are composed of system characteristics, user characteristics, organizational characteristics, and other factors. The specific model is shown in Figure 1.

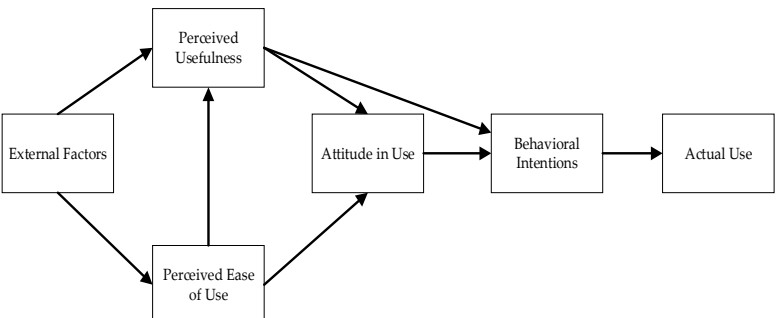

**Figure 1.** TAM model.

Since the TAM model was proposed, scholars have always had different opinions on the relationship between perceived usefulness, perceived ease of use, intention to use, and attitudes toward the use of new systems. For example, Venkatesh and Davis (1996) stated that usage attitudes were only users' preferences for information technology, which was reflected in their emotions, and could not fully convey the influence of useful and easy-to-use perceptions on behavioral intentions. In addition, the TAM model and expanded TAM were used to explain the acceptance of new technologies and new systems. In previous studies of extended TAM, there have been few secondary constructions abstracted from perceived usefulness and perceived ease of use to explore new technologies. This

indicates that the TAM model has many shortcomings in identifying the reasons for people's acceptance of new information systems and needs to be revised according to the specific situation.

Venkatesh and Davis et al. [12] proposed the behavioral model Unified Theory of Acceptance and Use of Technology (UTAUT) by integrating eight behavioral theoretical models which were Theory of Reasoned Action (TRA), Technology Acceptance Model (TAM), Model of PC Utilization (MPCU), Theory of Planned Behavior (TPB), Innovation Diffusion Theory (IDT), Social Cognition Theory (SCT), composite TAM and TPB model, and Motivation Model. They extracted four factors from them that influence users' acceptance motivation, namely, effort expectation, performance expectation, social impact, and contributing factor. They also extracted four moderating variables which were age, gender, experience, and voluntary. The Theory of Reasoned Action (TRA) shows that the personal perception and prevailing perceptions of the society in which one lives are important determinants of a person's attitudes and values. Individual attitudes and values determine a person's motivation to adopt a particular behavior, and motivation ultimately determines whether a behavior is adopted by a person. The Theory of Planned Behavior (TPB) shows that factors influencing behavioral willingness include behavioral beliefs, which have a potential influence on individual attitudes to perform the behavior, and normative beliefs, which are subjective norms that influence individual behavior. This means that information influences willingness and subsequent behavior through attitudes and subjective norms [13]. Innovation Diffusion Theory (IDT) is defined as a rational contemplation that seeks to clarify how, why, and to what degree new ideas and technologies are being spread [14]. Based on the recognition that individuals have subjective motivation, the Social Cognitive Theory (SCT) systematically reveals the process of generating individual behavior from individual cognition. In SCT, human behavior is extensively motivated and regulated by the ongoing exercise of self-influence. SCT conceives individuals as being goal-directed and actively engaged in developing thought processes and behaviors to meet their goals. It highlights the interaction between personal goals, cognition, and contextual factors in regulating motivated behavior [15].

The UTAUT model extracts the important factors that can predict people's use of a particular technology. Due to the integration of various theories and models, the explanatory effect is better and more realistic in predicting individuals' acceptance behaviors of information technology compared to the TAM model [12]. In addition, the model achieves the highest explanatory validity for usage behavior. Therefore, it is widely used in many fields such as e-commerce and information technology, and the validity of the UTAUT model is about 10% higher than TAM in explaining individual behavior [16].

For example, Hoque et al. [17] studied the key factors influencing elderly users' intention to adopt and use the mHealth services by developing a theoretical model based on UTAUT model. Akinnuwesi et al. [18] investigated factors affecting users' intention to use biometric technology (BT) in a developing country based on the modified version of the UTAUT model. Moreover, Alalwan et al. [19] explained the key factors influencing Jordanian customers' intention and adoption of Internet banking by using the extended UTAUT model.

The UTAUT behavioral model is shown in Figure 2.

Do all of these variables have a significant effect on farmers' adoption intention? Based on prior knowledge, all survey areas in this study had network coverage and all villagers in the area had access to the network. There was no influence of the contributing factor, so the factor was deleted and replaced by "personal innovation". In addition, because most pig farmers were 40–50 years old in the pre-investigation, the moderating variables of the original model were deleted and an innovative UTAUT model was constructed as the research method. How do these factors affect the willingness of farmers' intention to adopt IoT traceability technology? What is the relationship between them? Considering the factors that influence pig farmers' adoption willingness of IoT traceability technology are complex and diverse, it was assumed that the adoption willingness is influenced by

a combination of performance expectancy, effort expectancy, social influence, personal innovation, and perceived risk. Accordingly, a hypothetical model of pig farmers' adoption willingness of IoT traceability technology is proposed, as shown in Figure 3.

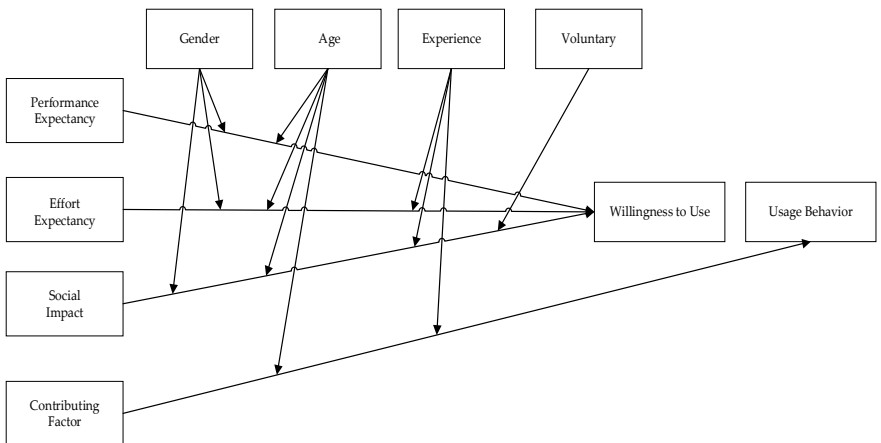

**Figure 2.** Unified Theory of Acceptance and Use of Technology (UTAUT).

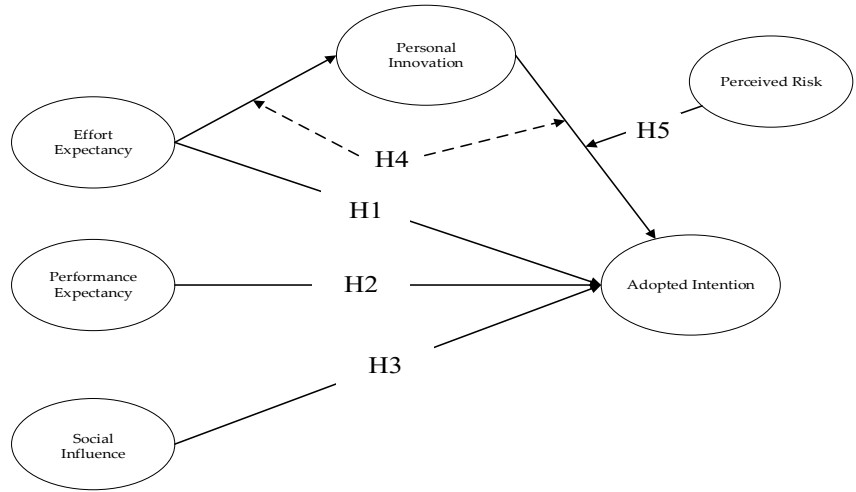

**Figure 3.** Latent variables and related hypotheses.

### 2.2. Research Hypotheses

In this study, effort expectancy refers to the ease of use and whether farmers perceive the IoT traceability technology to be simple to master. Knutsen's study found that effort expectancy had a significant impact on the adoption and use of emerging technologies. A study provided evidence that the more the system was perceived as effortless, the more likely it would be adopted by the individuals [20]. Therefore, this paper proposes the following hypothesis.

**Hypothesis 1 (H1).** *Farmers' effort expectations have a positive effect on the adoption willingness of IoT traceability technology.*

In this study, performance expectancy represents the extent to which farmers subjectively believe that the adoption of IoT traceability technology will bring improvements to their farming and marketing process. When performance expectancy is at a high level, users have a positive attitude towards using the system. For agriculture, studies found the importance of performance expectancy on the intentions of farmers to adopt mobile-based technologies for agricultural information. This implies that farmers' intentions to use apps

will be strengthened if they believe that the apps will result in greater performance in their daily agricultural activities [21]. Internet of Things is one of the most popular subjects today, where sensors and smart devices facilitate the provision of information and communication. The supporting role of IoT traceability technology on farming influences the adoption of IoT technology in agricultural quality traceability systems [22]. Accordingly, the following hypothesis is proposed.

**Hypothesis 2 (H2).** *Performance expectations of farmers have a positive effect on adoption willingness of IoT traceability technology.*

Social influence refers to the extent to which farmers perceive whether others think they should adopt IoT traceability technology. The positive impact of the social influence on the behavioral intentions sheds light on the convincing effect of the farmers' coworkers and farmhands in persuading them to use IoT in farming [23]. The more the farmer's neighbors, relatives, friends, and village cadres support the farmer in adopting the new technology, the higher the farmer's willingness is. In this study, if the number of relatives and friends around the surveyed farmers adopt IoT traceability technology is in a large proportion or recommend them to adopt the technology, it will naturally have a positive impact on farmers' adoption willingness. Therefore, the following hypothesis is proposed in this study.

**Hypothesis 3 (H3).** *Social influence has a positive impact on farmers' adoption willingness of IoT traceability technology.*

Personal innovation refers to the degree of farmers' personal acceptance of new things and it was found that personal innovation affects users' adoption willingness of a new technology. In this study, personal innovation represents the personal characteristics of the farmers in terms of actively exploring new and unknown things and the stronger personal innovation of farmers, the higher degree of initiative in exploring new things, which implies the stronger initiative in understanding and adopting IoT traceability technology. Dewi et al. [24] proposed that personal innovation had a crucial role in innovation adoption and also had a strong and direct effect on consumers' decision to adopt new technology because individuals became aware of new technology based on personal traits such as personal innovation. It can be concluded that the more innovative the consumers are, the higher the behavioral intention will be. Therefore, when the degree of initiative to explore something new is higher, the willingness of farmers to adopt IoT traceability technology will be stronger. Meanwhile, in the process of field research, the author found that farmers were more likely to take the initiative to learn about the IoT traceability technology when they perceived that the technology was easy to grasp. For example, after the team members explained the operation related to the technology, farmers showed higher acceptance level of the technology than before, which implied they showed strong personal innovation. Therefore, the following hypothesis is proposed in this paper.

**Hypothesis 4 (H4).** *Personal innovation is a mediating variable between farmers' effort expectations and adoption willingness.*

Perceived risk explains the extent to which individual farmers believe that there is a potential for adverse consequences from using IoT traceability technology networks. In this study, perceived risk represents the potential dangers that farmers perceive in using IoT traceability technology, such as the farming costs and the threats of personal information. In this paper, the latent variable of perceived risk is extracted by combining perceived risk with personal innovation. Wu et al. [25] classified perceived risk into four aspects, which were technology, function, behavior, and economy, and found that perceived risk negatively affected users' adoption willingness. At the same time, the higher the farmers'

perceived risk to new technology, the lower their acceptance of the technology. Therefore, this study puts forward the following assumption.

**Hypothesis 5 (H5).** *The mediating effect of perceived risk on farmers' personal innovation acts as a moderator.*

## 3. Materials and Methods

### 3.1. Data Preparation

The data used in this paper were obtained from field research in Bailiang Village, Shuangzhao Office, Qinhan New City, Xixian New District, Shaanxi Province; Podi Village, Junma Town, Liquan County, Xianyang City, Shaanxi Province; and Xinfeng Town, Lintong Area, Xi'an City, Shaanxi Province. Researchers randomly selected 90 pig farmers within each sample from July to October 2020 to form a data sample of 270 pig farmers. In order to make the researched farmers understand the content of the questionnaire more specifically, on the one hand, several trainings were given to the participants of the research. On the other hand, a video explanation was provided for farmers to understand the meaning of the terminologies such as IoT traceability technology and questions in the questionnaire. A total of 270 questionnaires were distributed, excluding some questionnaires with incomplete or wrong information. Finally, 264 valid questionnaires were obtained with an efficiency rate of 97%.

### 3.2. Variable Settings and Descriptive Statistics

Firstly, the variable indicators affecting adoption willingness were constructed according to previous studies, then the content of the questionnaire items was adjusted according to the pre-investigation, and the specific content of the formal questionnaire was determined. The participants of the pre-investigation were pig farmers in Bailiang Village, Shuangzhao Office, Qinhan New City, Xixian New Area, Shaanxi Province. The pre-investigation was conducted to test whether the questionnaire scale was applicable to the study of pig farmers' adoption willingness of IoT traceability technology, in which 49 questionnaires were collected. After that, researchers made the item analysis and exploratory factor analysis on the collected data. According to the results of the test, the items with factor loading less than 0.5 were excluded, and finally six groups of 46 items were obtained for formal research. The Likert scale method was used for the measurement of this paper, with values 1 to 5 corresponding to "strongly disagree", "somewhat disagree", "neutral", "somewhat agree", and "strongly agree", respectively.

#### 3.2.1. Dependent Variable

The Internet of Things (IoT) is a dynamic global network infrastructure with self-organization capabilities based on standard and interoperable communication protocols, in which virtual "things" have identities, physical properties, virtual characteristics, and intelligent interfaces, and are integrated seamlessly with information networks [26]. In 2007, the first traceability system in China with Universal Signage System began to be piloted in Carrefour Supermarket in Beijing. Professionals pointed out that consumers could scan the barcode or QR code on the outer package of the food bought in this supermarket with their smart phones. They could promptly find out all the information about the place of production, production date, supplier, and production raw materials of the food. Thus, the traceability of food safety was carried out in this place. If the food was found to have safety problems, consumers could quickly get the traceability information of the food. Food safety sectors could identify and deal with the food in time to reduce unnecessary losses at the same time.

For pig breeding, IoT traceability technology mainly refers to the application of a pig breeding traceability management system by placing ID cards on piglets. The Radio Frequency Identification (RFID) technology is used to scan the electronic tag to store all the data of the breeding stages from breeding to birth, including management information such as medicine and vaccination. In addition, the IoT traceability technology may detect and control the environmental conditions in the breeding process in real time, such as the temperature, humidity, ventilation conditions of piggery, and the amount of cleaning and maintenance in the processing workshop [27].

The measure of pig farmers' adoption willingness was divided into two indicators as follows.

AI1: I am very willing to adopt the existing IoT traceability technology; and

AI2: I am willing to take the initiative to understand and consider adopting IoT traceability technology if there is an opportunity.

Both indicators are based on the Likert scale with values 1 to 5 corresponding to "strongly disagree", "somewhat disagree", "neutral", "somewhat agree", and "strongly agree".

### 3.2.2. Independent Variables

(1)  Performance expectancy. The variable is described by the following three indicators. Participants believe that the use of IoT traceability technology can largely improve the efficiency of pig farming. Participants believe that pig sales can be helped to a great extent through the use of IoT traceability technology. Participants believe that the use of the Internet is a great improvement to life.

(2)  Effort expectancy. The variable is described by the following three indicators. After learning about IoT traceability technology, participants think it is easy to master. If there is a simpler IoT traceability technology, participants are very likely to use it. Participants find the Internet is very convenient.

(3)  Social influence. The variable is described by the following four indicators. Participants have heard many people talk about IoT traceability technology. Participants have been recommended IoT traceability technology by slaughterhouse staff, wholesalers, and consumers many times. Participants have been recommended using IoT traceability technology by many family members and friends. Participants have been recommended to use IoT traceability technology by many people from governmental regulatory departments and quarantine departments.

### 3.2.3. Mediating Variable: Personal Innovation

This variable is described by the following two indicators. Participants are willing to take the initiative to learn about new food safety technologies. Participants strongly believe in the policy information promoted in the village.

### 3.2.4. Moderating Variable: Perceived Risk

This variable is described by the following two indicators. Participants are very worried that the IoT traceability technology will cause loss to their profit. Participants are very distrustful of the detection capability of the IoT traceability technology.

The indicators for the specific questions are shown in Table 1.

**Table 1.** Scale indicators.

| Variables | Indicators | Indicator Content |
|---|---|---|
| Performance expectancy (PE) | PE1 | The use of IoT traceability technology can greatly improve the efficiency of pig farming. |
| | PE2 | The use of IoT traceability technology can largely help to complete the pig sales. |
| | PE3 | The use of the Internet can make a big difference to life. |
| Effort expectancy (EE) | EE1 | After learning about IoT traceability technology, I think IoT traceability technology is easy to master. |
| | EE2 | If there is a simpler IoT traceability technology, I am very likely to use it. |
| | EE3 | I find the Internet is very convenient. |
| Social impact (SI) | SI1 | I have heard a lot of people talk about IoT traceability technology. |
| | SI2 | Slaughterhouse staff, wholesalers, and consumers have recommended I use IoT traceability technology many times. |
| | SI3 | Many family members and friends have recommended I use IoT traceability technology. |
| | SI4 | People from governmental regulatory departments and quarantine departments have recommended I use IoT traceability technology. |
| Personal innovation (PI) | PI1 | I am very willing to take the initiative to learn about new food safety technologies. |
| | PI2 | I strongly believe in the policy information promoted by the village. |
| Perceived risk (PR) | PR1 | I am very worried about the loss of my profits from using IoT traceability technology. |
| | PR2 | I am very distrustful of the detection capabilities of IoT traceability technology. |
| Adoption intention (AI) | AI1 | I am very willing to adopt the existing IoT traceability technology. |
| | AI2 | If given the opportunity, I would like to learn about and consider adopting IoT traceability technology. |

### 3.2.5. Descriptive Statistical Analysis of Variables

Table 2 summarizes the basic information of the respondents.

**Table 2.** Descriptive statistics of the variables influencing the adoption willingness of IoT traceability technology.

| Statistical Characteristics | Classification Indicators | Number of People | Share (%) |
|---|---|---|---|
| Gender | Male | 207 | 78.41% |
| | Female | 57 | 21.59% |
| Age | 1 = 20~35 years old | 9 | 3.41% |
| | 2 = 35~50 years old | 113 | 42.81% |
| | 3 = 50~65 years old | 128 | 48.48% |
| | 4 = 65~80 years old | 14 | 5.30% |
| Education level | 1 = Never went to school | 22 | 8.34% |
| | 2 = Primary school and below | 67 | 25.38% |
| | 3 = Junior high school | 120 | 45.45% |
| | 4 = High school/secondary vocational school | 49 | 18.56% |
| | 5 = Bachelor/higher vocational school | 6 | 2.27% |
| Internet usage | Yes | 221 | 83.71% |
| | No | 43 | 16.29% |

Firstly, the proportion of men in the total number of the respondents is larger than that of women, which is mainly due to the influence of pig farming environment and the fact that most jobs are manual labor, which requires the help of men.

Secondly, in the age distribution, the proportion of farmers under 35 years old is relatively low, which is mainly because this group of farmers has less experience in keeping pigs and prefer to go out to work, fewer farmers in this age group are engaged in pig farming. The largest number of respondents, between the ages of 35 and 65 years old, accounts for 91%. This may be because this group of farmers are more experienced and

adaptable to the environment and more willing to engage in pig farming. Therefore, the number of farmers in this age range accounts for the largest percentage of respondents.

Thirdly, in terms of the education level of the respondents, the largest proportion of farmers with junior high school education level and below is over 80%. This indicates that most pig farmers are not highly educated and have limited ability to accept new technology and knowledge.

Fourthly, in terms of Internet use, the vast majority of farmers use the Internet. This is related to the fact that the Internet has become very popular in rural areas in recent years. Through the above analyses, we can find that most of the researched subjects were men, who were older, less educated, and more likely to use the Internet. The above characteristics are consistent with the basic situation of rural pig farmers at present.

### 3.3. Methods

The moderated mediation model implies that the independent variable X influences the dependent variable Y through the mediating variable M, and the mediation process (X→M→Y) is moderated by the moderating variable Z [28]. The existing moderated mediation effect test methods are based on multiple linear regression analysis of the explicit variables [29]. The most important shortcoming of the multiple linear regression analysis of mediating and moderating effects is the assumption that all variables are measured without measurement error, which results in an underestimation of the mediating and moderating effects. The biggest advantage of establishing the Structural Equation Model (SEM) for the analysis of moderated mediation effect is that it is a better way to set latent variables, effectively control measurement errors, and accurately estimate the values of mediating and moderating effects.

Although the analysis of moderated mediation effects based on the Structural Equation Model has obvious advantages, the application of this method is not common in practice [30]. Wang [31] suggested that one possible reason for this was that the current analysis of moderated mediation effects based on structural equation model requires the use of product-indicator approaches, which required the use of product indicator as the index for the potential moderator. The product-indicator approaches had two major shortcomings. First, the generation of product indicators was complex, and there were multiple strategies for generating indicators, which were not easy to be mastered by general researchers. Different strategies for generating product indicators might produce different parameter estimates, which might cause confusion for the researchers in understanding and interpreting. Second, the product terms were non-normally distributed, which made the parameter estimation results based on the assumption of normal distribution produce bias and had problems of robustness. Fang et al. [32] pointed out that a feasible solution was to use the Latent Moderate Structural equations (LMS) method to perform the analysis of moderated mediation effects based on SEM because the LMS method did not require the use of the product indicator and avoided the problem of the product indicator. They also explored how to use the LMS method to perform the analysis of moderated mediation effects based on SEM. Facing the analysis of moderated mediation effects based on SEM tasks, Fang et al. [32] summarized a set of analysis processes as follows.

(1)  Judge whether the baseline SEM model is acceptable or not; if not, the analysis is finished, otherwise go to Step 2.
(2)  Judge whether the moderated mediation effects based on SEM model is acceptable or not; if not, the analysis is finished, otherwise go to Step 3.
(3)  Use the coefficient multiplication method to analyze the moderated mediation effects, if the bootstrap confidence interval excludes 0, it means that the moderated mediation effects are significant, as shown in Figure 4.

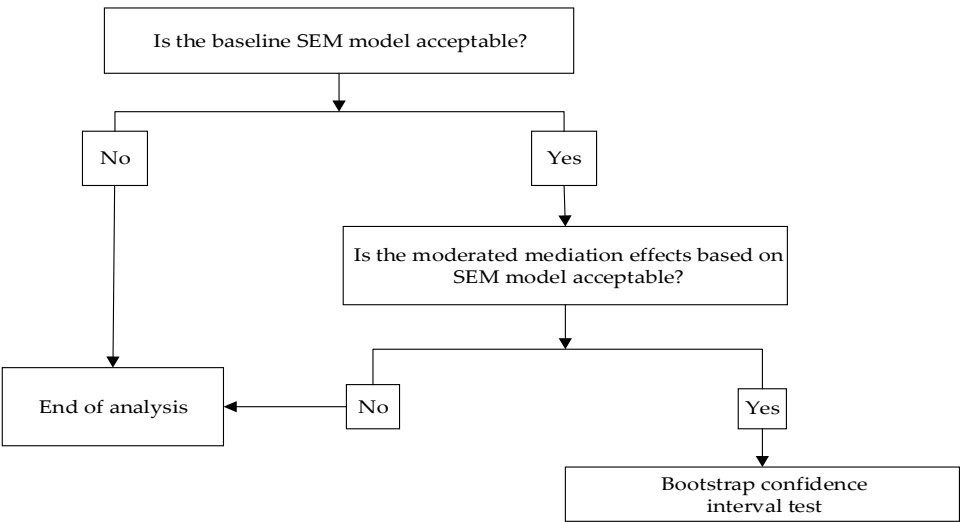

**Figure 4.** Flow chart of the analysis of moderated mediation effects based on SEM.

## 4. Structural Equation Model and Result Analysis

### 4.1. Data Quality Analysis

#### 4.1.1. Reliability and Validity Analysis

Cronbach's alpha coefficient is the most common test index for internal consistency reliability; Kaiser–Meyer–Olkin Measure of Sampling Adequacy (KMO) test and Bartlett's spherical test are common validity tests. In this paper, the above tests were conducted by SPSS Statistics 24.0 software (International Business Machines Corporation, New York, NY, USA) to verify the suitability of the data for factor analysis. The Cronbach's alpha coefficients for each dimension are shown in Table 3; it can be seen that the Cronbach's alpha coefficients for each latent variable were greater than 0.7. This indicates that the scale has high reliability, dependability, and stability.

**Table 3.** Reliability test.

| Dimension | Variables | Cronbach's Alpha |
|---|---|---|
| Performance expectancy | PE1 PE2 PE3 | 0.871 |
| Effort expectancy | EE1 EE2 EE3 | 0.791 |
| Social impact | SI1 SI2 SI3 SI4 | 0.756 |
| Personal innovation | PI1 PI2 | 0.865 |
| Perceived risk | PR1 PR2 | 0.836 |
| Adoption intention | AI1 AI2 | 0.800 |

In addition, the results of KMO and Bartlett's spherical test for the 16 measures of adoption willingness of IoT traceability technology in pig farming show that the KMO value is 0.807. This indicates that the scale data has good validity. In the Bartlett's spherical test value, the approximate chi-square value was 2025.370 and the significance level was 0.000. Thus, it was appropriate to conduct factor analysis on the data.

### 4.1.2. Model Simulation Test

The following indexes were used in this study to measure the fitting of the measurement model: Chi-square/degree of freedom, the Root Mean Square Error of Approximation (RMSEA), Comparative Fit Index (CFI), Non-Normed Fit Index (NNFI), Tucker–Lewis Index (TLI), Incremental Fit Index (IFI), and Standardized Residual Mean Square (SRMR). It can be seen in Table 4 that the results of the model tests in this study satisfied the range of judgmental criteria values.

**Table 4.** Results of overall model fitness index values.

| Indicators | $\chi^2$ | df | $\chi^2$/df | RMSEA | CFI | NNFI | TLI | IFI | SRMR |
|---|---|---|---|---|---|---|---|---|---|
| Criteria values for judgement | | | <3 | <0.10 | >0.9 | >0.9 | >0.9 | >0.9 | <0.1 |
| Results | 152.338 | 69 | 2.208 | 0.068 | 0.952 | 0.916 | 0.936 | 0.952 | 0.028 |

### 4.2. Analysis and Discussion of the Model

In this paper, the maximum likelihood estimate method was used to estimate the model parameters by using AMOS23 software (International Business Machines Corporation, New York, NY, USA), and the parameter estimation model is shown in Figure 5.

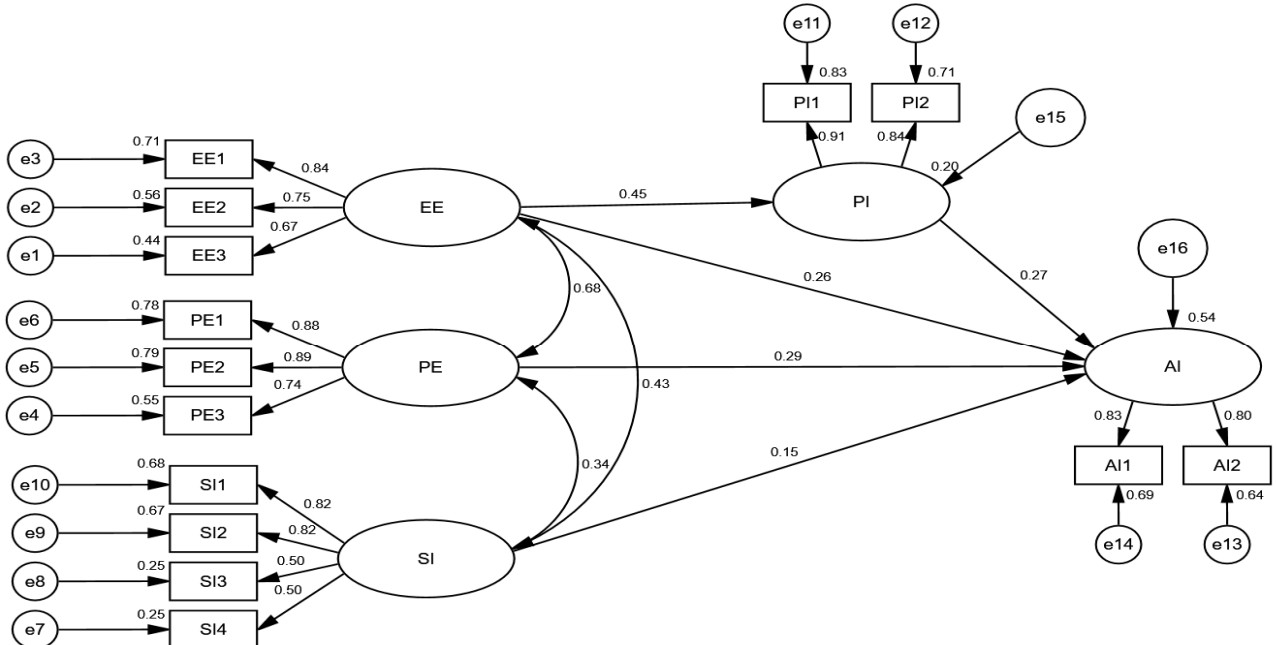

**Figure 5.** Structural equation model and normalized coefficients.

### 4.2.1. Hypothesis Test and Results

From the model estimation and hypothesis results (Table 5) as well as the structural equations and standardized path coefficients (Figure 6), it can be seen that the path coefficient of farmers' effort expectancy on their willingness to adopt IoT traceability technology was 0.262 and passed the significance test at the 5% level, indicating that farmers' effort expectancy significantly and positively affects their willingness to adopt IoT traceability technology. This means the hypothesis (H1) that farmers' effort expectancy has a positive influence on their willingness to adopt IoT traceability technology holds. The path coefficient of farmers' performance expectancy to farmers' adoption willingness of IoT traceability technology was 0.290 and passed the significance test at the 1% level, indicating that farmers' performance expectancy has a significant positive influence on their adoption willingness of IoT traceability technology. This means the hypothesis (H2) that farmers' performance expectancy has a positive influence on the adoption willingness of IoT trace-

ability technology holds. The influence path coefficient of farmers' social environment on their adoption willingness was 0.146 and passed the significance test of 5%, indicating that the social impact has a significant positive influence on farmers' adoption willingness of IoT traceability technology. This means the hypothesis (H3) that social impact has a positive influence on farmers' willingness to adopt IoT traceability technology holds.

**Table 5.** Model estimation and hypothesis results.

| Paths | Directions | Standardized Coefficients | S.E. | C.R. | P | Test Results |
|---|---|---|---|---|---|---|
| EE→AI | + | 0.262 | 0.054 | 2.485 | 0.013 | H1 is established. |
| PE→AI | + | 0.290 | 0.051 | 3.199 | 0.001 | H2 is established. |
| SI→AI | + | 0.146 | 0.121 | 1.984 | 0.047 | H3 is established. |

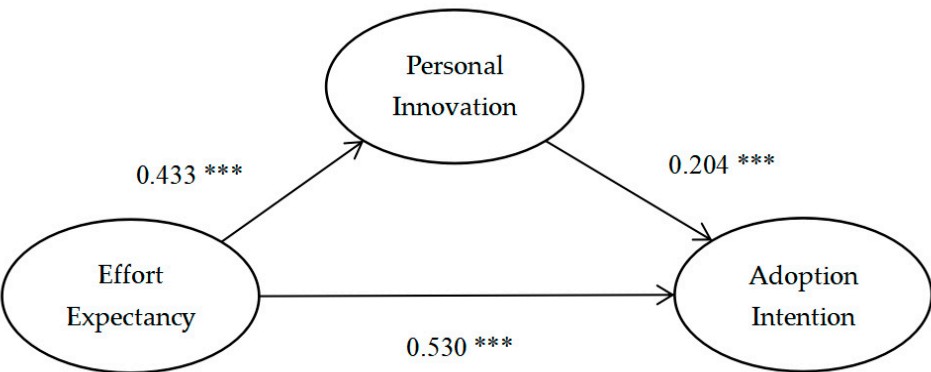

**Figure 6.** Mediation effect test. *** It indicates that the impact of the path is extremely significant.

### 4.2.2. Test of Mediating Effect

In this paper, we used the bootstrapping method to test the mediating effect of farmers' personal innovation by setting the sample number to 2000 and the confidence interval as 95%, and judged the existence of the mediating effect by the test results. The results show (Figure 6) that the indirect effect of personal innovation on farmers' effort expectancy and adoption willingness was 0.088 (0.433 × 0.204), and the path coefficient was significant at the 1% level. The bias-corrected confidence interval of the mediating effect was [0.056, 0.22], indicating that the lower limit of the indirect effect of farmers' effort expectancy on the adoption willingness of IoT traceability technology was 0.056 and the upper limit was 0.22, and the confidence interval excluded 0. Therefore, personal innovation has a significant mediating effect on farmers' effort expectancy and adoption willingness, and the hypothesis (H4) that personal innovation is a mediating variable of farmers' effort expectancy and adoption intention is valid.

### 4.2.3. Moderated Mediating Effect Test

The mediating effect of personal innovation was tested above. Next, the mediator variable and regulated variable were included in the model at the same time, and the analysis of moderated mediating effect was carried out by the coefficient multiplication method. It can be seen from the path test results in Table 6 that when farmers were at high risk perception, the moderating effect of the path of perceived risk on farmers' personal innovation and adoption willingness of IoT traceability technology was significant when farmers with a path coefficient of 0.158 and a bootstrap confidence interval of [0.09, 0.268] at the 95% level excluding 0, indicating that the moderating effect was significant. When farmers were at moderate risk perception, the moderating effect of the path of perceived risk on farmers' personal innovation and adoption willingness was significant with a path coefficient of 0.088, and the bootstrap confidence interval was [0.026, 0.17] at the 95% level excluding 0, indicating that the moderated mediating effect was significant. When farmers were at low risk perception, the path moderation effect of perceived risk on farmers'

personal innovation and adoption willingness was not significant, and the bootstrap confidence interval was [−0.086, 0.137] at the 95% level including 0, which indicates that the mediating effect with moderation was not significant. Therefore, the hypothesis (H5) is valid. The above results indicate that with the increase of the moderating variable (perceived risk), the mediating effect of farmers' personal innovation on the adoption willingness of IoT traceability technology increases significantly, which implies that the moderating variable significantly moderates the degree of the mediating effect.

**Table 6.** Tests of moderated mediating effects based on moderating path analysis.

| Regulated Variables | Path Coefficients | Bias-Corrected 95% IC | | |
|---|---|---|---|---|
| | | Lower | Upper | P |
| High risk perception | 0.158 | 0.09 | 0.268 | 0 |
| Moderate risk perception | 0.088 | 0.026 | 0.17 | 0.009 |
| Low risk perception | 0.019 | −0.086 | 0.137 | 0.643 |

*4.3. Analysis of Estimation Results*

4.3.1. Effort Expectancy

Effort expectancy is the degree to which farmers personally perceive whether IoT is easy to use and simple to master. Effort expectancy has a significant positive influence on farmers' adoption willingness of IoT traceability technology. It implies that the degree to which farmers subjectively perceive the technology to be easy to use in relation to their own reality through appropriate explanation by investigators under the existing degree of IoT technology popularity. Due to the intelligence of IOT devices and the learning ability of farmers, we found in the actual survey that when "IoT traceability technology" was first mentioned, most farmers had a low willingness to adopt IoT traceability technology because they had never heard of it and did not know anything about it. However, after listening to the brief explanation and examples given by investigators, farmers showed higher enthusiasm and willingness to adopt than before, which also conformed to the test results of the data. This was because when farmers felt that the technology was easy to master, the learning cost in practical application might decrease. From a profit perspective, farmers were more willing to adopt it. Therefore, the higher the effort expectancy of farmers for the technology, the stronger the willingness to adopt IoT traceability technology. In other words, effort expectancy positively influences farmers' adoption willingness of IoT traceability technology.

4.3.2. Performance Expectancy

In this study, performance expectancy represents the extent to which farmers subjectively believe that the adoption of IoT traceability technology will have benefits for their farming and marketing process. Performance expectancy has a significant positive effect on farmers' willingness to adopt IoT traceability technology. In the survey, it was found that most of the farmers' questions about IoT traceability technology focused on the cost and consumer acceptance. High performance expectancy means that farmers subjectively predict that IoT traceability technology will bring higher profits to their farming and marketing process. Except for a small number of family pig farmers, most farmers raise pigs to maintain living expenses, so as long as the technology can increase the existing profit amount, farmers will show a high willingness to adopt IoT traceability technology. This means that performance expectancy positively influences farmers' adoption willingness of IoT traceability technology.

4.3.3. Social Influence

Social influence refers to the extent to which farmers personally perceive whether others think they should adopt IoT traceability technology. For example, if someone among the farmers' relatives, friends, neighbors, intermediaries, or consumers they usually come into contact with recommended the farmers use IoT traceability technology or had used

IoT traceability technology, then the social influence was at a high level. The more people recommended or used it, the greater the social influence. In the survey, it was found that most farmers usually use cell phones, computers, and other electronic products to access the Internet mainly for entertainment, so it is impossible for them to learn and understand IoT traceability technology from the Internet. News and mass media reports on IoT traceability technology are rare, so farmers' knowledge of IoT traceability technology mostly comes from the surrounding environment. When the social influence is greater, farmers will have more expectation and trust in the convenience or profit brought by IoT traceability technology. Moreover, due to the influence of conformity psychology, a higher level of social influence will also have a positive impact on farmers' adoption of IoT traceability technology. This means that social influence significantly and positively affects farmers' adoption of IoT traceability technology.

### 4.3.4. Personal Innovation

Personal innovation refers to the degree of farmers' personal acceptance of new things. When a farmer is willing to take the initiative to understand and learn more about emerging technologies, the degree of his personal innovation is higher. In addition, when a technology is easier for a farmer to master, which implies the farmer's effort expectancy is higher, he tends to be more willing to take the initiative to learn about it. Therefore, effort expectancy has a significant positive effect on personal innovation. Moreover, when the farmer's personal innovation is stronger, his enthusiasm for learning emerging technologies including IoT traceability technology is stronger, so his adoption willingness is higher. This means that farmers' personal innovation has a significant positive influence on their adoption of IoT traceability technologies. In summary, farmers' effort expectancy significantly and positively influences farmers' adoption willingness through personal innovation as a mediating variable.

### 4.3.5. Perceived Risk

Perceived risk refers to the extent to which farmers personally believe that the use of IoT traceability technologies will likely have adverse consequences. The results of the data analysis show that the mediating effect of perceived risk on farmers' personal innovation and willingness to adopt plays a significant positive moderating role. In contrast, general research suggests that perceived risk negatively affects adoption willingness, which implies that the lower the perceived risk, the more significant the impact of farmers' personal innovation on adoption willingness should be. Wu et al. (2010) divided perceived risk into four aspects, which were technology, function, behavior, and economy, and found that perceived risk significantly and negatively affects users' willingness to use. However, through the data analysis, we found that the greater the perceived risk of farmers, the greater the impact of their personal innovation on adoption willingness. In fact, this is because the stronger perceived risk means that farmers are more active in learning the new technology and tend to make a prudent risk judgment after understanding a new technology, rather than unconditionally trusting new technology in order to complete the questionnaire. The author found in the communication with farmers during the field survey that the more skeptical farmers were about IoT traceability technology, the more willing they were to actively ask the investigators questions about the specific operation of IoT traceability technology and showed stronger initiative and enthusiasm. This means that personal innovation will significantly and positively affect their willingness to adopt IoT traceability technology. Thus, the effect of perceived risk on personal innovation has a significant positive moderating effect.

## 5. Conclusions

There have been a series of food safety incidents that have brought great harm to people's health in China and the application of IoT traceability technology is conducive to ensuring food quality and safety, improving the public's awareness of traceability products, and promoting the steady development of social economy. However, farmers are affected by their own conditions, social environment, cultural beliefs, and other factors. Most of them have low willingness to adopt new technologies.

It is necessary to identify and classify the influencing factors that affect farmers' adoption of IoT traceability technology, which will have a positive impact on resilience in agri-food supply chains and sustainability. The higher willingness of farmers to adopt IoT traceability technology means higher agility in the agri-food supply chains. Supply chain agility positively contributes to supply chain resilience [33]. In other words, with more transparent information exchanges and better joint collaboration, supply chain members are able to prepare for, adapt to, and recover from the risks better, which means that supply chain agility positively contributes to supply chain resilience. Moreover, the willingness of farmers to adopt new technologies explored in this paper contributes to the application of circular principles in supply chain systems. Applying the principles of circularity to the supply chains allows new rules to be established with suppliers and customers. It increases the number of actors with an active role in greener operations. A long-term partnership between customers and suppliers is fundamental to achieve social and environmental solutions [34]. Finally, the digital technology studied in this paper improves the sustainability of agricultural production. Digital technologies increase the operational efficiency through the accessibility and collection of process data in real time, the management of energy and resource consumption, and knowledge of the entire life cycle (design, manufacturing, distribution, maintenance, and use) with the potential to eliminate discontinuities and inefficiencies [35].

In fact, both resilience and sustainability are viewed as distinct concepts, but are positively correlated [1]. On the one hand, resilience has a positive impact on sustainability. Giudice et al. [36] mentioned that achieving a sustainable food system means "increasing or maintaining agricultural yields and efficiency while decreasing the environmental burden on biodiversity, soils, water and air." Klumpp et al. [37] also pointed out that the efficiency reductions after IT disruptions occur at different levels and for diverse reasons, and successful preparation and contingency management could support improvements. Moreover, the pandemic period has caused severe socio-economic damage, but it is accompanied by environmental deterioration that can also affect economic opportunities and social equity. In the face of this double risk, future generations are ready to be resilient and make their contribution not only on the consumption side but also through their inclusion in companies by bringing green and circular principles with them [38]. These examples all show the positive effect of resilience on sustainability to some extent. On the other hand, sustainability has a positive effect on resilience. A profound and holistic discussion is emerging around the question of how sustainable the present food system is and how prepared it is to face the kind of shock posed by the COVID-19 pandemic. Fabio Giudice et al. pointed out that circular practices improved resilience of the entire value chain (from production to consumption and post-consumption) through the introduction of localized supply chains.

Therefore, based on research data from 264 pig farmers in Shaanxi Province, the innovative UTAUT model was established. The researchers verified the research hypothesis through empirical analysis and analyzed the factors that influence pig farmers' adoption willingness of traceability technology.

The contributions in this article can be differentiated between theoretical and practical contributions.

## 5.1. Theoretical Contribution

First of all, the previous research on the IoT traceability technology was mainly about the innovation of the technology and the combination with blockchain. These studies regarded pig farmers' adoption intention as an exogenous variable and ignored its endogeneity. This article filled the gap in the research of IoT traceability technology.

Secondly, in the past, the Structural Equation Model was mostly used to study the adoption intention, ignoring the relationship between independent variables. The Structural Equation Model including the intermediate variable and the latent variable established in this study makes up for the shortcomings of previous studies and puts forward an impact path that is more in line with the actual situation.

## 5.2. Practical Contribution

First, the higher the expectancy of farmers' efforts, the stronger their willingness to adopt the IoT traceability technology. This shows that farmers are more willing to adopt the technology when the actual operation of the technology is easier than the farming methods they use at present. In fact, Fedushko et al. [39] pointed out that the developed machine learning model made a difference to improve transaction tracing. This helped identify errors, enhance operations, data pipelines to make a project requirement precise, identify use-cases, and apply monitoring for project improvement. Moreover, continuous real-time monitoring combined with machine learning for a certain industrial operational use-case allowed a system to increase availability which was one of the factors that led to higher user satisfaction levels. Second, when farmers predict that the technology will bring higher profits, their willingness to adopt it is stronger. Puriwat et al. [19] pointed out that when people knew that social media was useful for business purposes and using social media as an alternative business platform would enhance their business performance, they would be more willing to adopt social media for business purposes. Third, farmers tend to show higher adoption willingness of the technology when they are surrounded by people who have recommended the technology to them, especially when they have already used it for pig farming. The more people recommend and use the technology, the higher the farmers' adoption willingness. Wissal et al. [40] pointed out that one of the strongest predictors of patients' behavioral intention to use connected devices in healthcare was social influence. Health was a personal matter. However, as people were often not experts in many health-related issues, they were easily impacted by the important others in their social groups, such as their family physicians. Fourth, when the IoT traceability technology is simpler and easier for farmers, and farmers are more active in understanding it, they are more likely to adopt the technology. It means that personal innovation as a mediating variable of farmers' effort expectancy and adoption willingness has a significant positive effect on the results. Fifth, the personal characteristics of farmers when faced with a new technology have a significant effect on their adoption willingness, mainly because farmers are skeptical about new technology. This means that the farmers are more motivated to learn, thus the effect of personal innovation on adoption willingness is more significant.

Based on the above research conclusions, the following policy inspirations are obtained. First, the improvement of IoT traceability technology at the technical level plays an important role in its popularization, so the relevant departments should increase the investment in the research of this technology and strive to make the actual operation of farmers using this technology easier than the existing traditional farming methods. Second, for the farmers who have adopted the technology, government departments should give appropriate subsidies, which not only plays a role in the protection of farmers after the adoption of new technology risk, but also encourages more farmers to adopt the technology. Third, the government, village committees, and other relevant departments should increase the publicity of IoT traceability technology. These departments should not only carry out technical promotion and publicity work on the farmers themselves, but also on other environmental factors that may potentially affect the farmers, such as other villagers

and intermediaries, so as to help improve the adoption willingness of IoT traceability technology as a whole.

### 5.3. Limitation and Future Recommendation

In the existing literature, we found that most researchers studied the farmers' adoption intention as an exogenous variable, so this paper tried to take the adoption intention as an important endogenous variable in the distribution and promotion of new technologies. However, in practice, we clearly perceived that the willingness to adopt new technologies was only one of many endogenous variables that had not been studied. There were many factors affecting it besides those listed in this paper. In other words, our research on the popularity of IoT traceability technology and the influencing factors of adoption intention are not complete.

In future research, we should continue to explore the endogenous variables that affect the adoption of new technologies. Exploring the impact of these factors on adoption intention by developing different theoretical models to improve the integration and resilience of the supply chains will contribute to the sustainability of agricultural development. In addition, the results of this article showed that the mediating effect of perceived risk on farmers' personal innovation and willingness to adopt played a significant positive moderating role. As the result is inconsistent with previous studies, we will continue further discussion regarding whether perceived risk has a significant impact on users in different variables (e.g., gender, education, usage experience, etc.) adopting new technologies.

**Author Contributions:** Conceptualization, R.S. and S.Z.; methodology, R.S. and J.H.; software, R.S.; formal analysis, R.S.; investigation, R.S. and S.Z.; writing—original draft preparation, R.S.; writing—review and editing, T.W. and J.R. (Junhu Ruan); supervision, J.R. (Junyong Ruan). All authors have read and agreed to the published version of the manuscript.

**Funding:** This research received no external funding.

**Institutional Review Board Statement:** Ethical review and approval were waived for this study, due to the fact that we used anonymous data that was not retraceable to individuals at any time.

**Informed Consent Statement:** Patient consent was waived due to the fact that we used anonymous data that was not retraceable to individuals at any time.

**Data Availability Statement:** The data are not publicly available due to confidentiality reasons.

**Conflicts of Interest:** The authors declare no conflict of interest.

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
