# Peer review of "Willingness and Influencing Factors of Pig Farmers to Adopt Internet of Things Technology in Food Traceability"

_sustainability, doi:10.3390/su13168861_

Round 1

Reviewer 1 Report

The topic of food supply chain traceability is not new in the literature; however, the authors of this paper provide the reader with solid justifications to bring out the innovative contribution of their research.

The manuscript follows a rigorous approach but requires some improvements before its publication.

In the abstract, although quite extensive, not highlighted the literature gap that justifies this study.

To correlate the two knowledge GAPs highlighted by the authors, with the theoretical model and the final results, it would be desirable to formulate one or more research questions that would facilitate the reading of the article.

A theoretical model with empirical validation, such as the one proposed by the authors (UTAUT) requires a stronger literature base. Most importantly, an in-depth examination of the endogenous factors that hinder technology adoption is lacking. In this sense, the literature can be extended by showing the relationship between sustainability and resilience in agri-food supply chains (https://doi.org/10.1007/s11356-020-11130-2) as well as in other sectors (https://doi.org/10.3390/su13116130 and  https://doi.org/10.3390/su13042052). 

Both the theoretical and analytical models are described in detail and clearly, as well as the results obtained are well discussed.

Both organizational (for farmers) and policy implications are appreciated in the conclusions, however, the theoretical implications that this study brings to the literature on the topic are missing. 

I encourage the authors to make the suggested improvements in order to make this manuscript suitable for publication.

Reviewer 2 Report

This paper deals with new ideas to solve the problem of smart production and quality traceability of meat foods and offers new ideas for the formation of safe and high-quality markets for meat products using IoT in Food Traceability provides.

The abstract of the paper should be improved. The aim of the paper should be better stated. The abstract should be a single paragraph and should follow the style of structured abstracts but without headings:

1) Background: Place the question addressed in a broad context and highlight the purpose of the study;

2) Methods: Describe briefly the main methods or treatments applied. Include any relevant preregistration numbers, and species and strains of any animals used.

3) Results: Summarize the article's main findings,

4) Conclusion: Indicate the main conclusions or interpretations.

Authors should take into account more previous works (e.g. theoretical, conceptual, and empirical reviews) published in the literature. Authors should discuss the results and how they can be interpreted from the perspective of previously published studies. 

An approach to using operational Intelligence to solve industrial technology projects problems is very crucial for today’s IT is suggested in this reference: Fedushko, S.; Ustyianovych, T.; Gregus, M. Real-Time High-Load Infrastructure Transaction Status Output Prediction Using Operational Intelligence and Big Data Technologies. Electronics 20209, 668. https://doi.org/10.3390/electronics9040668.

The authors should reorganize the conclusion section because the contents of the conclusion are not clear. The conclusion should include the following contents as background, research objective, experiment result, finding and future research, and limitations.

I suggest adding a concluding paragraph with that, how these main findings of the paper address the challenge of sustainability.

Thank you for a good job.

Round 2

Reviewer 1 Report

Dear Authors 

I have attentively read the latest version of your manuscript appreciating the considerable improvements you have made following the reviewers' advice. Therefore, I now consider your article suitable for publication in this journal.

Congratulations!
